

# The effect of rotenone contamination on high-resolution mitochondrial respiration experiments

Dale F. Taylor[1], Jia Li[1], Nicholas J. Saner[1], Jia Wei[1], Xu Yan[1], Elizabeth G. Reisman[2], Hanzhe Li[1], Matthew J-C Lee[1,3], Navabeh Zare[1], Andrew Garnham[1], Jujiao Kuang[1] and David J. Bishop[1]

[1] Institute for Health and Sport, Victoria University, Melbourne, Victoria, Australia
[2] Mary MacKillop Institute for Health Research, Australian Catholic University, Melbourne, Victoria, Australia
[3] Institute for Physical Activity and Nutrition, School of Exercise and Nutrition Sciences, Deakin University, Geelong, Victoria, Australia

## ABSTRACT

**Background**. High-resolution respirometry is commonly used in skeletal muscle research and exercise science to measure mitochondrial respiratory function in both permeabilized muscle fibers and isolated mitochondria. Due to the low throughput and high cost of the most used respirometer, the Oroboros 2k (O2k), multiple experiments are often conducted within the same chamber in short succession. Despite this, no methodological consideration has been given for the potential contamination of inhibitors, used to investigate the contribution of specific complexes within the electron transport chain, between experiments.

**Methods**. We first assessed the potential effect of inhibitor contamination on mitochondrial respiration experiments by evaluating the ability of the currently recommended wash protocol to remove rotenone and compared its efficacy against a simplified wash protocol of sequential rinses. Secondly, we assessed the potential effect of inhibitor contamination on mitochondrial respiration measured before and after a single session of high-intensity interval exercise, with and without the use of rotenone between experiments.

**Results**. The currently recommended protocol for washing chambers was insufficient for removing rotenone. Following exercise, a decrease in mitochondrial respiration was observed exclusively in chambers exposed to rotenone between experiments.

**Discussion**. Our findings highlight an important methodological consideration regarding the measurement of mitochondrial respiratory function using high-resolution respirometry, with inhibitor contamination potentially affecting the conclusions derived from experiments conducted in close succession. Future studies investigating mitochondrial respiratory function should assess the necessity of using inhibitors such as rotenone, ensure thorough wash procedures between experiments, and explicitly report the washing protocols used.

Corresponding author
Jujiao Kuang, Jujiao.Kuang@vu.edu.au

## INTRODUCTION

Mitochondria are double-membrane-bound organelles present in most eukaryotic cells and are the primary site of ATP synthesis through oxidative phosphorylation (OXPHOS) (*Anderson et al., 2019*). In response to exercise, mitochondria are capable of rapid modification through a process termed mitochondrial biogenesis—defined as the synthesis of new components of the mitochondrial reticulum (*Irrcher et al., 2003*). This process is particularly important for skeletal muscle, where increased energy demand leads to changes in its energy-generating capacity—also known as mitochondrial respiratory function. Due to the importance of energy generation in health and performance, several techniques have been developed to measure mitochondrial respiratory function in skeletal muscle *via* the measurement of oxygen consumption (*Djafarzadeh & Jakob, 2017*).

To assess mitochondrial respiratory function, high-resolution respirometers detect oxygen consumption using an oxygen-sensing electrode in an enclosed chamber. Through the stepwise addition of specific substrates, uncouplers, and inhibitors, the contribution of different components of the electron transport system (ETS) to OXPHOS can be assessed. When analyzing mitochondrial respiratory function in skeletal muscle, fibers are either mechanically separated using forceps to increase their surface area for interaction with the chamber media and permeabilizing chemicals, or, alternatively, a crude mitochondrial extract is isolated through homogenization and stepwise centrifugation.

Using commercially available respirometers, such as the Oxygraph-2k Respirometer (Oroboros Instruments, Innsbruck, Austria), most studies have reported varying levels of improvement in the different states of mitochondrial respiration following both moderate-intensity and high-intensity exercise training (*Chrois et al., 2020*; *Granata et al., 2021*; *Granata et al., 2016b*; *MacInnis et al., 2017*). Conversely, some (*Layec et al., 2018*; *Lewis et al., 2021*; *Trewin et al., 2018*) but not all (*Newsom et al., 2021*; *Tonkonogi et al., 1999*), studies have identified decreases in mitochondrial respiratory function following a single session of exercise; this was despite no detected changes in markers of mitochondrial content (*e.g.*, citrate synthase activity). However, the use of different protocols for assessing mitochondrial respiration, regarding both the preparation of samples and the substrate-uncoupler-inhibitor titration (SUIT) protocols used, makes comparing the results between studies difficult (*Liao et al., 2020*). Furthermore, the reproducibility of exercise-induced changes in mitochondrial respiration is likely impacted by the small sample sizes of many studies (*Granata, Jamnick & Bishop, 2018*) and the technical variability of high-resolution respirometry (*Cardinale et al., 2018*; *Jacques et al., 2020*; *Kuang et al., 2022*).

A potential confounding factor that has not been addressed is the risk of inhibitor contamination on mitochondrial respiratory function, particularly when experiments are conducted within a few hours on the same machine—a scenario that is often unavoidable in many laboratories. Despite the manufacturer's recommended wash protocol for the Oxygraph-2k using multiple distilled water and ethanol washes (*Schmitt, 2025*), there is evidence to suggest certain chemicals, such as Complex I inhibitor rotenone (ROT), used to measure electron flow through only Complex II, and Complex III inhibitor antimycin A (AMA), required to measure non-mitochondrial oxygen consumption, remain present

after washing with this protocol (*Djafarzadeh & Jakob, 2017*). If inhibitor contamination is suspected, Oroboros recommends incubating mitochondrial-rich isolates or tissues in the chamber for 30 min (*Schmitt, 2025*). However, this procedure is rarely performed and not reported in studies that have documented a post-exercise decrease in mitochondrial respiration. Even trace amounts of these inhibitors could affect subsequent respiration experiments, potentially leading to inaccurate results or misleading conclusions. This study, therefore, aimed to investigate the effects of residual ROT contamination on successive experiments conducted within the same chambers, both in resting muscle and following exercise.

To assess the potential effect of ROT contamination on mitochondrial respiratory function, several experiments using permeabilized human muscle fibers were performed. Firstly, we assessed whether rinsing Oroboros chambers exposed to ROT with 70% (v/v) ethanol, 100% (v/v) ethanol, and distilled $H_2O$ is sufficient for cleaning. This experiment, composed of a rinsing protocol commonly used immediately prior to the first use of an Oroboros machine after removing the 70% (v/v) ethanol used for overnight storage, aimed to determine the significance of wash duration and establish a baseline for comparison with the recommended Oroboros cleaning procedure. Secondly, an assessment of whether the currently recommended wash protocol by Oroboros is sufficient to clean chambers exposed to ROT was performed. Thirdly, a trial with muscle biopsies collected prior to and 6 h following a high-intensity interval exercise session was performed. This third trial, which had half of the available chambers exposed to ROT following the initial biopsy experiment, was used to assess whether decreases in mass-specific respiration following exercise can be correctly attributed to the effects of exercise. A fourth trial, which included the assessment of mitochondrial respiratory function without the use of ROT, was used to confirm the effect of a single exercise session on mass-specific mitochondrial respiration. This fourth trial had muscle biopsies collected prior to, and 3 h following, a single session of high-intensity interval exercise.

## MATERIALS & METHODS

### Participants

Participants were provided with information regarding the requirements of these studies (Fig. 1), including the benefits and risks, before providing both verbal and written informed consent. All participants were free of musculoskeletal injuries, any condition or disease that affects cardiovascular function (*e.g.*, heart rhythm disturbance, elevated blood pressure, diabetes), and were not taking certain prescription medications (*e.g.*, beta-blockers, anti-arrhythmic drugs, statins, or insulin-sensitizing drugs).

**Experiments 1 & 2:** Muscle biopsies were obtained at rest from 12 healthy men and 4 healthy women (30.8 ± 6.1 y; 174.0 ± 10.3 cm; 76.6 ± 16.6 kg, mean ± SD) from several studies approved by the Human Research Ethics Committee at Victoria University (HRE22-057, HRE20-065, and HRE22-184).

**Experiment 3:** Muscle biopsies were obtained from five healthy men and women (26.3 ± 4.5 y; 167.8 ± 4.7 cm; 74.4 ± 15.8 kg, mean ± SD) from a study approved by the Human Research Ethics Committee at Victoria University (HRE20-212).
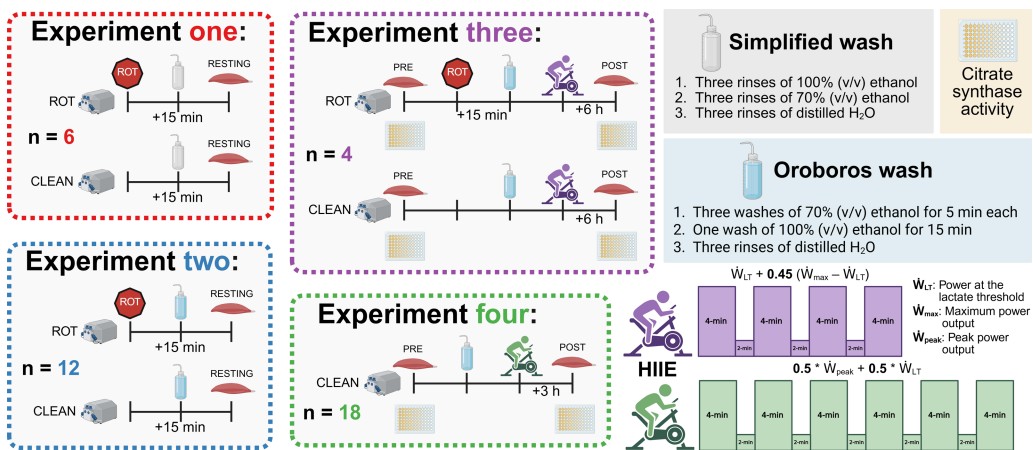

**Figure 1** **Experimental protocol.** ROT, rotenone; HIIE, high-intensity interval exercise. Created using Biorender.com.

**Experiment 4:** Muscle biopsies were obtained from 10 healthy men from a study approved by the Human Research Ethics Committee at Victoria University (HRE18-214). A full description of the recruitment and participant details has previously been described (*Li et al., 2022*).

## Muscle biopsies

All participants reported to the laboratory in a fasted and rested state, having not had food, alcohol, or completed exercise in the 12 h prior to the biopsy trials. Following the injection of local anesthesia (1% lidocaine), an experienced medical doctor took all muscle biopsies from the *vastus lateralis* muscle using a suction-modified Bergstrom needle (*Tarnopolsky et al., 2011*). From each biopsy, 15 to 20 mg of muscle was rapidly placed into an ice-cold biopsy preservation solution (BIOPS) for high-resolution mitochondrial respiration. For Experiments 3 and 4, excess muscle was immediately frozen in liquid nitrogen and stored for the measurement of citrate synthase activity (as a marker of mitochondrial content; (*Larsen et al., 2012*)).

## Exercise protocols

Both exercise trials (Experiments 3 and 4) used high-intensity interval exercise (HIIE) on a stationary bicycle. Similar HIIE protocols have been employed within 4- to 6-week training studies by our research group to significantly increase mitochondrial respiratory function (*Granata, Jamnick & Bishop, 2018*; *Granata et al., 2016a*).

**Experiment 3:** Following the first biopsy, the participants performed an exercise protocol consisting of four 4-min intervals, separated by 2 min of rest on a stationary bicycle (Velotron Cycle Ergometer; RacerMate Inc., Seattle, WA, USA). The intensity of each interval was set at the power at the lactate threshold ($\dot{W}_{LT}$) + 0.45 * (maximum power output ($\dot{W}_{max}$) − $\dot{W}_{LT}$), with $\dot{W}_{max}$ and $\dot{W}_{LT}$ determined using a ramped and graded exercise test, respectively, as previously performed by our group (*Jamnick et al., 2018*).

The second biopsy was taken 6 h following the completion of exercise (POST); this time point was selected due to the required time to complete both the pre-existing mitochondrial respiration SUIT protocol and the recommended Oroboros cleaning protocol (*Di Marcello et al., 2023*). Between biopsies, participants consumed a standardized diet of 2.2 g kg$^{-1}$ of bodyweight of carbohydrates, 0.7 g kg$^{-1}$ of bodyweight of protein, and 0.4 g kg$^{-1}$ of bodyweight of fat, which was spread across two meals eaten at approximately 0.5 and 3.5 h post exercise.

**Experiment 4**: Following the first biopsy, the participants performed an exercise protocol consisting of six 4-min intervals, separated by 2 min of rest, on a stationary bicycle (Velotron Cycle Ergometer; RacerMate Inc.). The intensity of each interval was set at 0.5 $^*$ peak power output ($\dot{W}_{peak}$) + 0.5 $^*$ $\dot{W}_{LT}$, with $\dot{W}_{LT}$ and $\dot{W}_{peak}$ determined using a graded exercise test as previously performed by our group (*Jamnick et al., 2018*). A second biopsy was taken 3 h following the completion of exercise (POST). A full description of the baseline testing, exercise protocol, and the between-biopsy conditions have previously been described (*Li et al., 2022*).

## Mitochondrial respiration
### Separation, permeabilization, and calibration
Immediately after the biopsy, each small section of muscle in BIOPS underwent gentle manual fiber separation on ice. The fibers were subsequently inspected under a microscope to ensure they were intact before undergoing incubation with gentle shaking for 20 min in ice-cold BIOPS solution containing 50 µg mL$^{-1}$ of saponin. After three 7-min washes in gently agitated ice-cold mitochondrial respiration medium (MiR05), the muscle samples were briefly blotted on filter paper and weighed. After weighing the fibers, each sample was split into either duplicate or triplicate, depending on sample and chamber availability, with 1.2–2.5 mg of intact muscle transferred into each high-resolution respirometer chamber (Oxygraph-2k; Oroboros) containing fresh MiR05.

Oxygen concentration (nmol mL$^{-1}$) and oxygen flux (pmol O$_2$ s$^{-1}$ mg$^{-1}$) were recorded using the Oroboros DatLab software. A background calibration for each Oroboros machine was performed within the 6 months prior to use. A correction for instrumental background oxygen flux, accounting for sensor oxygen consumption and oxygen diffusion between the medium and the chamber boundaries, was performed during the calibration process before each experiment. If air calibrations presented more than 5% deviation in the results, the membranes were changed, and a new background calibration was performed. Oxygenation by direct syringe injection of O$_2$ in the chamber was necessary to maintain O$_2$ levels between 200 and 550 nmol mL$^{-1}$, to prevent a low concentration of oxygen that would limit respiration.

### Wash protocols
**Simplified:** A simplified version of the Oroboros wash protocol was tested, replacing each 5-minute wash step with a rinse, and the results were compared with the recommended Oroboros protocol. The simplified wash protocol was composed of three washes of distilled H$_2$O, three washes of 70% (v/v) EtOH, and three washes of 100% (v/v) EtOH; each wash was completed in quick succession and consisted of completely filling the chamber prior

to subsequent siphoning. A final three washes of distilled $H_2O$ were completed prior to filling the chambers with MiR05 for the following experiment.

**Oroboros:** The recommended Oroboros protocol for washing Oxygraph-2k chambers involves soaking the chamber and stopper in 70% (v/v) EtOH for 5 min three times, siphoning off and replacing the 70% (v/v) EtOH each time (*Schmitt, 2025*). A final soak of the chamber and stopper using 100% (v/v) EtOH for 15 min is performed, with three washes using distilled $H_2O$ prior to the addition of MiR05 for the next experiment.

### SUIT protocols

**Experiment 1:** Prior to analyzing the permeabilized and separated muscle fibers, ROT (final concentration in the chamber of 0.5 µM) was titrated into fully operational chambers containing MiR05. Cleaning using the simplified protocol was performed when ROT had been within the chamber for 15 min, consistent with its use within a typical SUIT protocol (*Di Marcello et al., 2023*).

The SUIT protocol used with final chamber concentration in brackets was as follows: pyruvate (two mM) and malate (five mM) in the absence of adenylates were added for measurement of leak respiration with electron entry through Complex I ($CI_L$) (oxygen consumption through Complex I with substrates present but no ADP). A series of stepwise titrations of ADP (100, 250, 500, 750, 1,000, 2,000, 5,000, 7,500, 10,000 µM) was then added and the final concentration served as a measurement of peak oxidative phosphorylation (OXPHOS) respiratory function with electron input through complex I ($CI_P$) (oxygen consumption through Complex I with substrates and ADP present to allow oxidative phosphorylation). Following this, succinate (10 mM) was added for the measurement of OXPHOS activity with simultaneous electron supply through Complex I and complex II ($CI+II_P$) (oxygen consumption through Complex I and Complex II with substrates and ADP present to allow oxidative phosphorylation). Cytochrome c (10 µM) was then added to test for outer mitochondrial membrane integrity; an exclusion criterion was set such that if a chamber showed an increase in $O_2$ flux over 15% after the addition of cytochrome c it was to be discarded from analysis, this threshold was based on previous assessments into mitochondrial respiration variability (*Kuang et al., 2022*). This was followed by stepwise carbonyl cyanide-4-(trifluoromethoxy)phenylhydrazone titrations (FCCP, 0.5–1 µM) for measurement of electron transport system (ETS) capacity with convergent electron input through complex I and II ($CI+II_E$) (oxygen consumption without the limitations of ATP synthase or membrane potential of $CI+II_P$). This was followed by the addition of antimycin A (AMA, 2.5 µM), an inhibitor of complex III, to obtain a measurement of residual oxygen consumption that was subtracted from all other measurements to account for non-respiration oxygen consumption in the chamber. A summary of SUIT protocols for all experiments is provided in Fig. 2.

**Experiment 2:** The same SUIT protocol as Experiment 1 was performed. However, instead of the simplified wash protocol following the initial titration of ROT, the recommended Oroboros wash protocol was utilized.

**Experiment 3:** The same SUIT protocol as Experiment 1 & 2 was performed; however, no ROT was titrated prior to the initial (PRE) biopsy. Instead, at the end of the SUIT protocol

**Experiment one, two, three:**
- **CI<sub>L</sub>**: Mal + Pyr
- **CI<sub>P</sub>**: Mal+ Pyr + ADP
- **CI+II<sub>P</sub>**: Mal + Pyr + ADP + Suc
- **CI+II<sub>E</sub>**: Mal + Pyr + ADP + Suc + FCCP

**Membrane integrity**: < 15% change following Cyt c
**Non-mitochondrial O₂ consumption**: AMA

**Experiment four:**
- **ETF<sub>L</sub>**: Mal + Oct
- **ETF + CI<sub>P</sub>**: Mal + Oct + Pyr + ADP
- **ETF + CI+II<sub>P</sub>**: Mal + Oct + Pyr + ADP + Suc
- **ETF + CI+II<sub>E</sub>**: Mal + Oct + Pyr + ADP + Suc + FCCP

**Membrane integrity**: < 15% change following Cyt c
**Non-mitochondrial O₂ consumption**: AMA

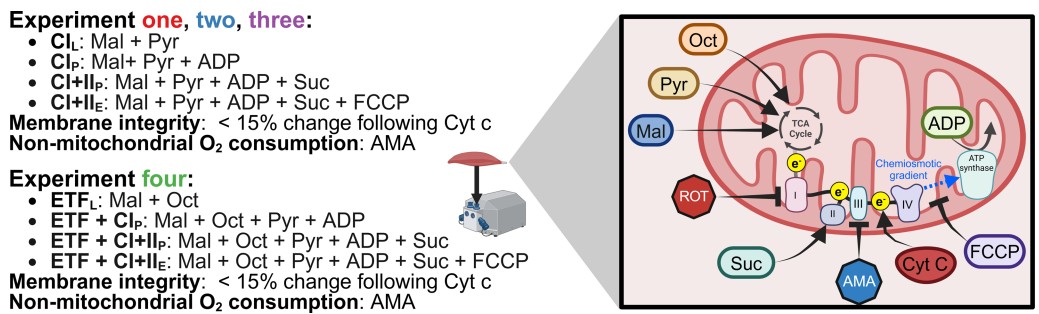

**Figure 2** **Substrate-uncoupler-inhibitor titration (SUIT) protocols used.** Sequence of substrates, uncouplers, and inhibitors used within each experiment. Pyr, pyruvate; Mal, malate; Oct, octanoyl-carnitine; ROT, rotenone; Suc, succinate; AMA, antimycin A; Cyt c, cytochrome c; FCCP, carbonyl cyanide 4-(trifluoromethoxy) phenyl- hydrazone; ADP, adenosine diphosphate; TCA, tricarboxylic cycle. Created using Biorender.com.

for the PRE samples, ROT (final concentration in the chamber of 0.5 µM) was titrated into half of the technical replicates. Cleaning using the recommended Oroboros protocol was performed when ROT had been within the chamber for at least 15 min (*Di Marcello et al., 2023*).

**Experiment 4**: The SUIT protocol used with final chamber concentration in brackets was as follows: octanoyl-carnitine (0.2 mM) and malate (two mM) in the absence of ADP was added to measure the leak respiration *via* electron transferring flavoprotein (ET$F_L$). A titration of $MgCl_2$ (three mM), ADP (5 mM), and pyruvate (five mM) were then added for measurement of respiration through ETF and complex I (ETF+CI$_P$). Followed by the addition of succinate (10 mM) for a measurement of peak OXPHOS respiratory function with simultaneous electron supply through ETF, complex I, and complex II (ETF + CI + II$_P$). cytochrome c (10 mM) was then added to check the integrity of mitochondrial outer membrane, followed by stepwise titrations carbonyl cyanide 4-(trifluoromethoxy) phenyl-hydrazone (FCCP) titrations to plateau (0.75–1.5 µM), for measurement of ETS capacity (ETF+CI+II$_E$) followed. Antimycin A (2.5 mM) was then added to measure residual oxygen consumption. Cleaning of the chambers between experiments occurred using the recommended Oroboros wash protocol.

## Citrate synthase activity assay

Citrate synthase (CS) activity was determined using small sections of frozen skeletal muscle homogenized to a concentration of six µg µL⁻¹ for Experiment 3 and 4 µg µL⁻¹ for Experiment 4 in ice-cold lysis buffer containing 50 mM Tris, 150 mM NaCl, 1% (v/v) Triton X-100, 1 mM EDTA, and 1:100 (v/v) protease inhibitor cocktail (Cell Signaling Technology, Danvers, MA, USA) at a pH of 7.4. In triplicates, five µL of muscle homogenate was analyzed using 40 µL of three mM acetyl-CoA, 25 µL of one mM 5,5′-dithiobis(2-nitrobenzoic acid) (DTNB), and 165 µL 100 mM Tris buffer (pH 8.3) kept at 30 °C. After the addition of 15 µL of 10 mM oxaloacetic acid, the plate was quickly placed in an xMark Microplate spectrophotometer (Bio-Rad, Hercules, CA, USA), and after 30 s of linear agitation, absorbance at 412 nm was recorded every 15 s for 3 min at 30 °C.

## Statistical analyses

The *stats* and *emmeans* packages in R were used for all statistical analyses, with the assumptions of a normal distribution and equal variance. A two-tailed paired student's *t*-test was used to assess the effect of rotenone on mitochondrial respiration in Experiments 1 and 2, and the effect of exercise in Experiment 4. A repeated measures two-way ANOVA was used to test the effect of exercise, as well as the interaction between exercise and rotenone in Experiment 3. Post-hoc pairwise comparisons between groups in Experiment 3 were performed using estimated marginal means without adjustment for multiple comparisons, due to the small sample size, which limited the power of more stringent correction methods. Plots were generated using the *ggplot2* package in R. Source data to verify this research is provided with this paper and the R script used for analysis is deposited on GitHub and available at https://doi.org/10.5281/zenodo.14862623. Significance was determined as a $p$-value $< 0.05$, and exact $p$-values were reported throughout. All values of mitochondrial respiration reported in the text are the mean difference between conditions or time points unless otherwise stated.

## RESULTS

### Rinsing does not completely remove rotenone following an experiment

The aim of Experiment 1 was to investigate whether a simplified wash, consisting of distilled $H_2O$, 70% (v/v) ethanol, and 100% (v/v) ethanol, was sufficient to remove rotenone from the chambers of an Oxygraph-2k between experiments conducted on permeabilized skeletal muscle fibers. There was no effect of ROT exposure on $CI_L$ ($p = 0.160$; Fig. 3A). Measurements of $CI_P$ ($-61.13$ pmol O2 s$^{-1}$ mg of wet weight$^{-1}$, $p = 0.001$; Fig. 3B), $CI+II_P$ ($-24.96$ pmol O2 s$^{-1}$ mg of wet weight$^{-1}$, $p = 0.037$; Fig. 3C), and $CI+II_E$ ($-34.90$ pmol O2 s$^{-1}$ mg of wet weight$^{-1}$, $p = 0.013$; Fig. 3D) were significantly lower in chambers that were exposed to ROT compared to those that were not ($n = 6$).

### The currently recommended washing protocol is also insufficient at removing ROT contamination

Given that the simplified protocol was insufficient, we next investigated in Experiment 2 whether the currently recommended Oroboros wash protocol can remove rotenone from the chambers of an Oxygraph-2k between successive experiments conducted on permeabilized skeletal muscle fibers. There was again no effect of ROT exposure on $CI_L$ ($p = 0.565$; Fig. 4A). However, a significantly lower $CI_P$ ($-9.99$ pmol $O_2$ s$^{-1}$ mg of wet weight$^{-1}$, $p = 0.026$, Fig. 4B), $CI+CII_P$ ($-11.07$ pmol $O_2$ s$^{-1}$ mg of wet weight$^{-1}$, $p = 0.048$; Fig. 4C), and $CI+CII_E$ ($-11.98$ pmol $O_2$ s$^{-1}$ mg of wet weight$^{-1}$, $p = 0.047$; Fig. 4D) was still detected for chambers that were exposed to ROT, compared to those that were not ($n = 12$).

### A decrease in respiration following exercise is only observed in chambers exposed to ROT

Several studies have reported a 15–40% decrease in the mass-specific respiration of permeabilized muscle fibers following a single session of exercise (*Layec et al., 2018*; *Lewis*

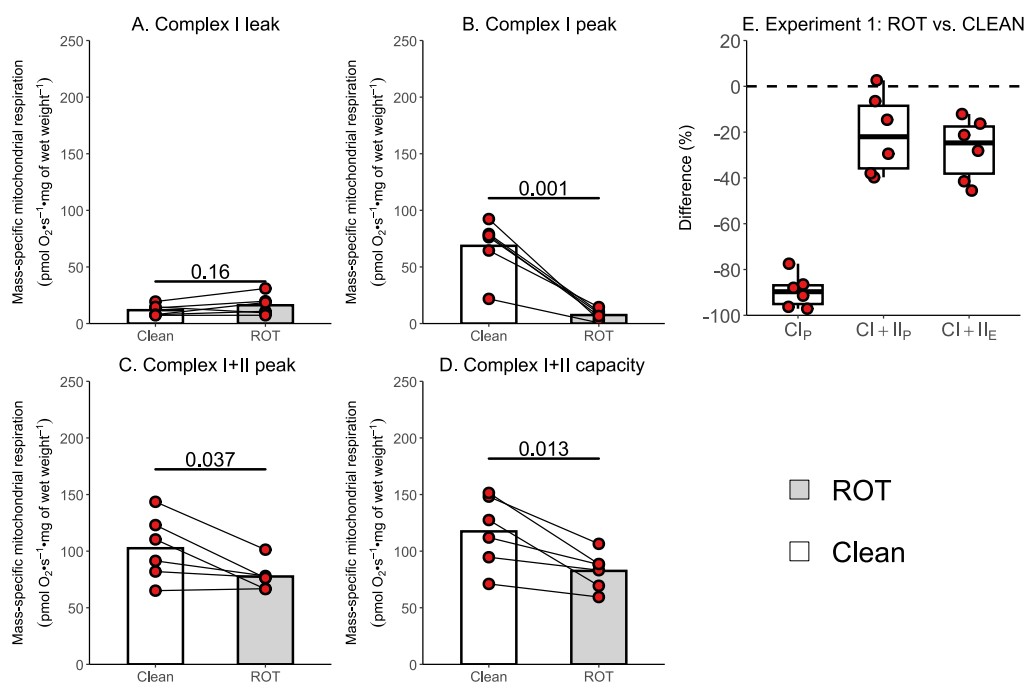

**Figure 3** **Experiment 1—Simplified wash of Oxygraph-2k chambers following exposure to rotenone (ROT).** (A) Leak respiration through complex I (complex I leak, CI$_L$). (B) Maximal coupled mitochondrial respiration through complex I (complex I peak, CI$_P$). (C) Maximal coupled mitochondrial respiration through complex I and II (complex I+II peak, CI+II$_P$). (D) Maximal uncoupled mitochondrial respiration through complex I and complex II (complex I+II capacity, CI+II$_E$). (E) Boxplot for difference between paired replicates not exposed to and exposed to ROT for CI$_P$, CI+II$_P$, CI+II$_E$. Box represents the interquartile range (IQR), the whiskers 1.5*IQR, and the central line the median value. Bars represent mean, unless otherwise stated.

*et al., 2021*; *Trewin et al., 2018*); however, an assessment of the potential contribution of inhibitor contamination to these results has not occurred. The aim of Experiment 3 was to investigate whether insufficient removal of rotenone might contribute to a reported decrease in mass-specific respiration in permeabilized skeletal muscle fibers following a single session of HIIE. Mass-specific mitochondrial respiration was significantly lowered following exercise in permeabilized human skeletal muscle when chambers were exposed to ROT ($n = 4$), but not in those that were not ($n = 4$).

Once again, no changes in CI$_L$ were observed following exercise, regardless of exposure to ROT (Fig. 5A). Although CI$_P$ ($-21.91$ pmol O$_2$ s$^{-1}$ mg of wet weight$^{-1}$, $p = 0.097$; Fig. 5B) and ($-21.81$ pmol O$_2$ s$^{-1}$ mg of wet weight$^{-1}$, $p = 0.076$; Fig. 5C) decreased within chambers exposed to ROT, they were below the threshold of significance. Following exercise, CI+II$_E$ ($-19.60$ pmol O$_2$ s$^{-1}$ mg of wet weight$^{-1}$, $p = 0.046$; Fig. 5D) was significantly decreased when chambers were previously exposed to ROT. In the chambers not exposed to ROT after the PRE sample, there was no significant decrease in CI$_P$, CI+II$_P$, and CI+II$_E$ following exercise ($p = 0.452$ to $0.967$). No changes in CS activity, a commonly

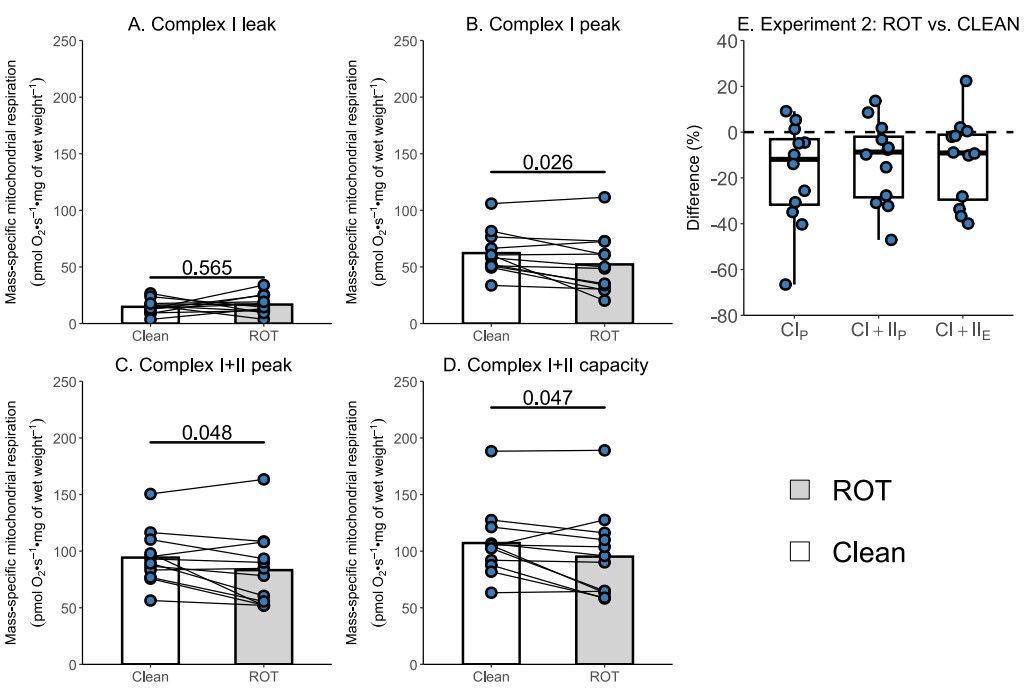

**Figure 4 Experiment 2—Recommended Oroboros wash protocol of Oxygraph-2k chambers following exposure to rotenone (ROT).** (A) Leak respiration through complex I (complex I leak, $CI_L$). (B) Maximal coupled mitochondrial respiration through complex I (complex I peak, $CI_P$). (C) Maximal coupled mitochondrial respiration through complex I and II (complex I+II peak, $CI+II_P$). (D) Maximal uncoupled mitochondrial respiration through complex I and complex II (complex I+II capacity, $CI+II_E$). (E) Boxplot for difference between paired replicates not exposed and exposed to ROT for $CI_P$, $CI+II_P$, $CI+II_E$. Box represents the interquartile range (IQR), the whiskers 1.5*IQR, and the central line the median value. Bars represent mean, unless otherwise stated.

used indirect marker of mitochondrial content (*Larsen et al., 2012*), were detected following exercise (Fig. 5F).

Due to the relatively small sample size used within Experiment 3, additional data generated within our research group using a similar experimental design, but without the use of rotenone, were used to further assess if a single session of high-intensity interval exercise is associated with a significant decrease in mass-specific respiration in the absence of ROT (Experiment 4). There was no significant effect of exercise on mass-specific mitochondrial respiration measurements $ETF+CI_L$ ($p = 0.172$; Fig. 6A), $ETF+CI_P$ ($-7.41$ pmol $O_2$ s$^{-1}$ mg of wet weight$^{-1}$, $p = 0.209$; Fig. 6B), $ETF+CI+II_P$ ($-10.06$ pmol $O_2$ s$^{-1}$ mg of wet weight$^{-1}$, $p = 0.257$; Fig. 6C), $ETF+CI+II_E$ ($-9.58$ pmol $O_2$ s$^{-1}$ mg of wet weight$^{-1}$, $p = 0.316$; Fig. 6D) ($n = 18$). Like Experiment 3, no changes in CS activity were detected following exercise (Fig. 6F).

## DISCUSSION

In this study, we investigated the potential effect of rotenone (ROT) contamination on mitochondrial respiration measurements in permeabilized human skeletal muscle

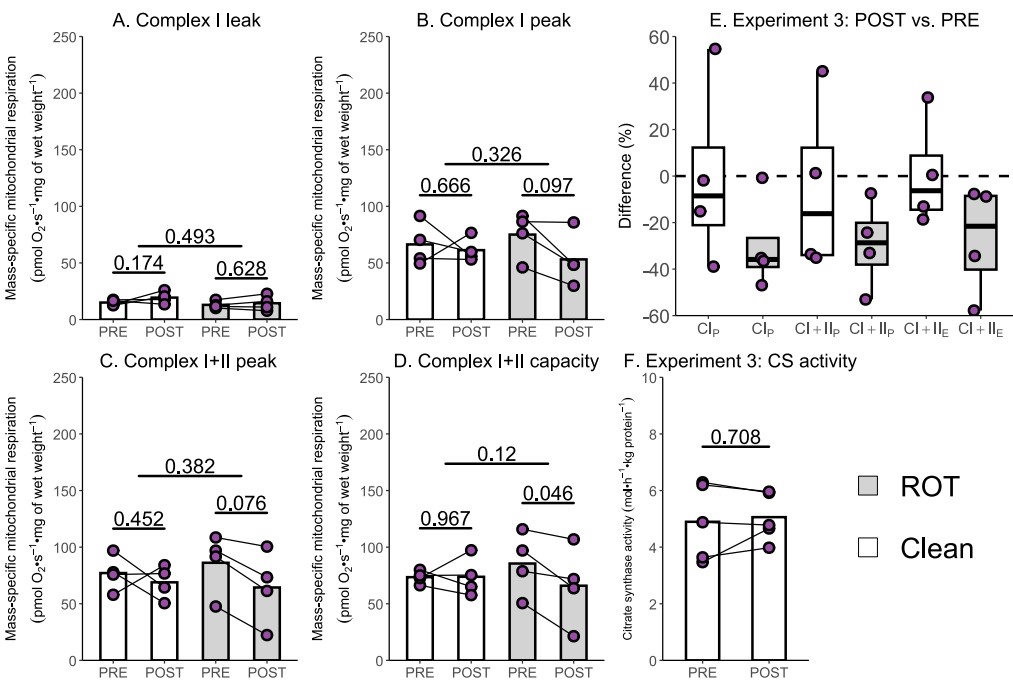

**Figure 5** **Experiment 3—The effect of high-intensity interval exercise and rotenone (ROT) contamination on mass-specific respiration.** (A) Leak respiration through complex I (complex I leak, $CI_L$). (B) Maximal coupled mitochondrial respiration through complex I (complex I peak, $CI_P$). (C) Maximal coupled mitochondrial respiration through complex I and II (complex I+II peak, $CI+II_P$). (D) Maximal uncoupled mitochondrial respiration through complex I and complex II (complex I+II capacity, $CI+II_E$). (E) Boxplot of difference between time points of paired replicates not exposed and exposed to ROT for $CI_P$, $CI+II_P$, $CI+II_E$. Box represents the interquartile range (IQR), the whiskers 1.5*IQR, and the central line the median value. (F) Citrate synthase (CS) activity in whole-muscle lysate. Bars represent mean unless otherwise stated.

fibers when experiments are completed in short succession using the same chambers. We demonstrated that when chambers are exposed to ROT, the wash protocol currently recommended by Oroboros is insufficient at cleaning the chamber and is associated with a significant decrease in commonly used measurements of mitochondrial respiratory function when experiments are completed within short succession.

In response to a single session of exercise, several studies have reported decreases in mitochondrial respiratory function (*Layec et al., 2018*; *Lewis et al., 2021*; *Trewin et al., 2018*), whereas others have observed no change (*Newsom et al., 2021*; *Tonkonogi et al., 1999*). While differences in exercise prescription or participant characteristics could contribute to these findings, the potential influence of commonly used inhibitors, such as ROT—which was present in all studies reporting a decrease—has not yet been evaluated. In Experiments 3 and 4, where repeated muscle biopsies following a single session of exercise were analyzed in the same respirometry chambers, a post-exercise decrease was only observed in those chambers previously exposed to ROT. For $CI+II_E$, which was significantly decreased in chambers exposed to ROT post-exercise in Experiment 3, a decrease of 30% compared to unaffected chambers is similar to previously reported decreases post-exercise

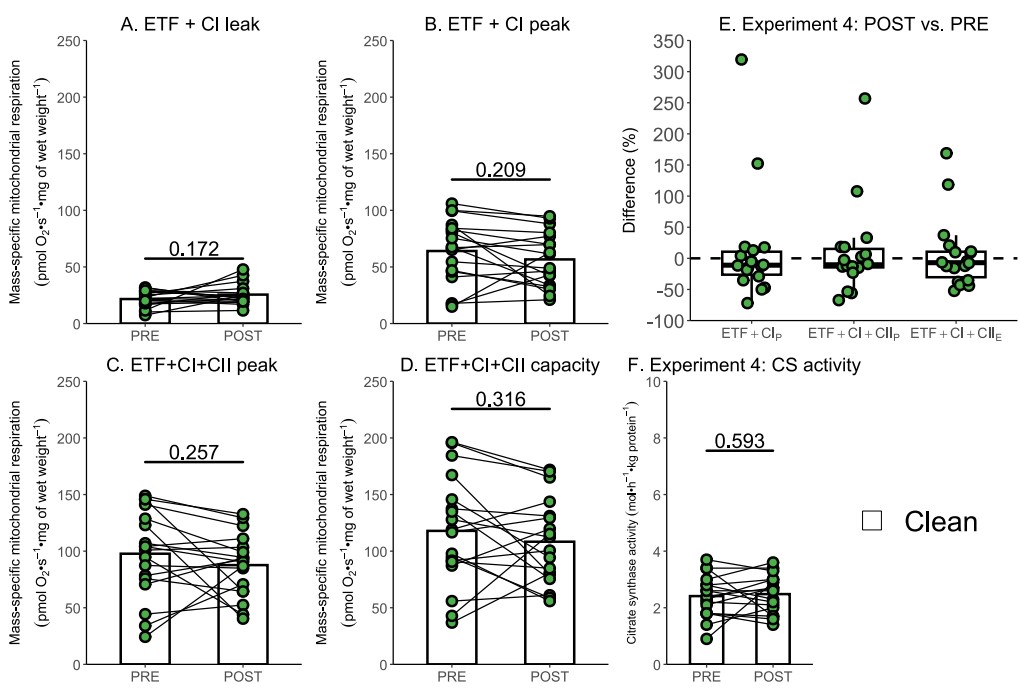

**Figure 6** **Experiment 4—The effect of high-intensity interval exercise on mass-specific respiration.** (A) Leak respiration through electron-transferring flavoprotein (ETF) and complex I (ETF + complex I leak, ETF+CI$_L$). (B) Maximal coupled mitochondrial respiration through ETF and complex I (ETF + complex I peak, ETF+CI$_P$). (C) Maximal coupled mitochondrial respiration through ETF, complex I, and complex II (ETF + complex I+II peak, ETF+CI+CII_P). (D) Maximal uncoupled mitochondrial respiration through ETF, complex I, and complex II (ETF + complex I+II capacity, ETF+CI+CII$_E$). (E) Boxplot of difference between time points for ETF+CI$_P$, ETF+CI+CII_P, ETF+CI+II$_E$. Box represents the interquartile range (IQR), the whiskers 1.5*IQR, and the central line the median value. (F) Citrate synthase (CS) activity in whole-muscle lysate. Bars represent mean, unless otherwise stated.

(ranging from 15–40% across different mitochondrial measurements) (*Layec et al., 2018*; *Lewis et al., 2021*; *Trewin et al., 2018*). Greater confidence in this conclusion can be gained by comparing these results against the much larger dataset of Experiment 4, with no significant change in mitochondrial respiration following high-intensity interval exercise in any measure using 18 replicates. While the number and intensity of the 4-min intervals were different between Experiment 3 and 4, both protocols had participants exercising at a high intensity (*Bishop et al., 2019*), with power above 75% of either $\dot{W}_{peak}$ or $\dot{W}_{max}$ (as determined by a graded exercise or ramped exercise test respectively) (*Li et al., 2022*). Although a slight, but not significant decrease, was observed for the measurements of CI$_P$ (8.3% and 11.6% respectively) and CI+II$_P$ (10.6% and 10.3% respectively) in chambers not exposed to ROT in Experiments 3 and 4, and CI+II$_E$ (8.1%) in Experiment 4, this may have arisen from AMA, which is required to measure non-mitochondrial respiration, also not being fully removed by the recommended Oroboros wash protocol.

To verify the impact of ROT contamination on mitochondrial respiratory function, we conducted two additional experiments. These included using a simplified wash protocol, composed of sequential rinses, and the recommended Oroboros wash protocol,

on chambers with and without exposure to ROT immediately prior to the respiration experiment. Consistent with its role in inhibiting complex I, the greatest percentage decreases in mitochondrial respiration were unsurprisingly observed in $CI_P$ for both the simplified and recommended Oroboros wash protocol. Although the Oroboros wash protocol (Experiment 2) did not result in as large a decrease in mitochondrial respiration as the simplified wash protocol (Experiment 1), significantly lower values for $CI_P$ (16% and 89% respectively), $CI+II_P$ (12% and 24% respectively), and $CI+II_E$ (11% and 30% respectively) were observed when chambers exposed to ROT were compared to the control chambers not exposed to ROT. The decreases observed with the use of the recommended Oroboros protocol were again consistent with previously reported reductions in mitochondrial respiratory function following exercise (*Layec et al., 2018*; *Lewis et al., 2021*; *Trewin et al., 2018*). Although Experiment 2 may theoretically overestimate the true effect in terms of reductions in mitochondrial respiration, it is unlikely that all available ROT would be sequestered by tissue at the concentrations typically used, particularly given that numerous prior studies have employed higher concentrations than those used in the present study, and that the reductions observed were greater in Experiment 3 than in Experiment 2. This further suggests that these decreases can be attributed to ROT contamination rather than any potential exercise-induced damage to skeletal muscle mitochondria.

Consistent with a decrease in mitochondrial respiratory function predominantly occurring due to a contamination of ROT, there was also no significant change in mitochondrial content (as assessed using CS activity (*Larsen et al., 2012*)) following exercise in Experiments 3 and 4. Although CS is not directly involved in oxidative phosphorylation, it does indicate that the upstream pathways involved in substrate metabolism are not perturbed by exercise. This provides confidence that the reductions in mitochondrial respiration observed in Experiment 3 result from impairments in the electron transport chain, rather than exercise-induced damage to the mitochondria.

Based on these experiments, if multiple respiration measurements are to be conducted using the same chambers on the same day, we recommend excluding ROT from SUIT protocols due to its ability to potentially impact subsequent experiments. If ROT or anything other inhibitor is to be used, it is essential to evaluate the effectiveness of any wash protocol employed to eliminate potential contamination between experiments. We further recommend that all manuscripts involving multiple respiration measurements within the same day include a detailed description of the wash protocol used between experimental runs to ensure the reproducibility of the results.

## CONCLUSIONS

In conclusion, we have highlighted a critical methodological consideration in commonly used SUIT protocols for high-resolution mitochondrial respiration. We recommend that future studies utilizing this technique should avoid using rotenone when assessing the mitochondrial respiratory function of permeabilized human skeletal muscle in successive experiments within the same day or employ a more thorough wash protocol than is

currently recommended. If rotenone or any other inhibitor is to be used and a separate wash protocol is devised, it should be tested for its effectiveness in removing contamination from previous experiments and reported as part of the experimental methods. To confirm the results within this study are applicable to all mitochondrial respiration experiments investigating changes from exercise and training, future studies with a similar methodology but with crude mitochondrial isolates and different exercise intensities may be beneficial.

## ACKNOWLEDGEMENTS

We thank all the participants for their time and effort in the studies. The authors would also like to thank the technical team at Victoria University for their assistance throughout the study.

### Funding

This work was supported by an Australian Research Council (ARC) Discovery Project grant (DP200103542), a National Health and Medical Research Council (NHMRC) grant (GNT2013427), and the Defence Science and Technology Group (DSTG) of the Department of Defence in Australia. The funders had no role in study design, data collection and analysis, decision to publish, or preparation of the manuscript.

### Grant Disclosures

The following grant information was disclosed by the authors:
Australian Research Council (ARC) Discovery Project: DP200103542.
A National Health and Medical Research Council (NHMRC): GNT2013427.
The Defence Science and Technology Group (DSTG) of the Department of Defence in Australia.

### Competing Interests

The authors declare there are no competing interests.

### Author Contributions

- Dale F. Taylor performed the experiments, analyzed the data, prepared figures and/or tables, authored or reviewed drafts of the article, and approved the final draft.
- Jia Li performed the experiments, authored or reviewed drafts of the article, and approved the final draft.
- Nicholas J. Saner performed the experiments, authored or reviewed drafts of the article, and approved the final draft.
- Jia Wei performed the experiments, authored or reviewed drafts of the article, and approved the final draft.
- Xu Yan performed the experiments, authored or reviewed drafts of the article, and approved the final draft.
- Elizabeth G. Reisman performed the experiments, authored or reviewed drafts of the article, and approved the final draft.

- Hanzhe Li performed the experiments, authored or reviewed drafts of the article, and approved the final draft.
- Matthew J-C Lee performed the experiments, authored or reviewed drafts of the article, and approved the final draft.
- Navabeh Zare performed the experiments, authored or reviewed drafts of the article, and approved the final draft.
- Andrew Garnham performed the experiments, authored or reviewed drafts of the article, and approved the final draft.
- Jujiao Kuang conceived and designed the experiments, authored or reviewed drafts of the article, and approved the final draft.
- David J. Bishop conceived and designed the experiments, authored or reviewed drafts of the article, and approved the final draft.

## Human Ethics

The following information was supplied relating to ethical approvals (*i.e.*, approving body and any reference numbers):

Ethics approval was granted by the Human Research Ethics Committee at Victoria University (HRE18-214, HRE22-057, HRE20-065, HRE20-212, and HRE22-184).

## Data Availability

Data and code are available at Zenodo:

Taylor, D. (2025). The effect of inhibitor contamination on subsequent high-resolution mitochondrial respiration experiments [Data set]. Zenodo. https://doi.org/10.5281/zenodo.14992000.

## Supplemental Information

Supplemental information for this article can be found online at http://dx.doi.org/10.7717/peerj.19879#supplemental-information.

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
