# Peer review of "The effect of rotenone contamination on high-resolution mitochondrial respiration experiments"

_PeerJ, doi:10.7717/peerj.19879_

## Round 0.1 · original submission · Major Revisions

Please address all reviewers' comments.

·

Basic reporting

The manuscript is clearly written, well-organised, and the figures, supplemental text are formatted in a professional manner. Referencing is sufficient and covers the points made.

Experimental design

The research reported is an original primary study that is meaningful and important to consider within the field of mitochondrial physiology. The experimental design is appropriate, and ethical standards have been met. The methodology is sufficiently detailed.

Validity of the findings

-

Additional comments

This is a well-written manuscript describing a careful set of experiments. The issue of rotenone contamination, due to a resistance to washing, is interesting and can have important implications for the community of biologists who are using respirometry to measure mitochondrial physiology. Many standard protocols that are widely used include rotenone, and as the authors mention, successive experiments on a single day are really normal.

The testing of a ‘simple’ wash protocol, showing a strong level of effect on CI, is a great response to the temptation to decrease wash protocol duration.

Given that sequential use of the respirometer is fairly unavoidable, there is the obvious question that arises. Do the authors have a recommended protocol for washing rotenone out of the chambers? The suggestion that wash protocols are reported is appropriate.

Suggestions to improve the manuscript
Could the word ‘rotenone’ be added to the title
A table of the samples used would help in understanding the study design of each experiment a little better, If it is easy, then this could be placed in supplemental information.

Experiment 1. For the pre-soak with rotenone, was this done in a fully operational chamber, lights and stirrers on? Light can help degrade rotenone.

Experiment 2. This yields more heterogeneous data, which is expected when using human tissue samples; some emphasis on this might be necessary for readers who normally work with cells or model organisms. Can any of the heterogeneity be accounted for by age or sex (12M and 4 F) differences? It would be good to know this had been considered, I don’t think it is necessary to include any new analyses (unless, of course, they change the conclusions somehow).

Experiment 3. The use of dot plots makes the data clear for the reader, again, though there is variability with data going in both directions. It would be interesting to have some further comments about the large variance in the clean chambers, particularly (Figure part E), POST vs PRE.

Experiment 4. This is the experiment done without rotenone. The dataset is much larger than for the previous experiments. Here, there isn’t a mix of sexes, this needs to be stated again in the text. The authors show that pre- and post-exercise, there is no significant difference in the
measures presented.

These are a good set of experiments that have been carried out to demonstrate the effect of residual rotenone on subsequent data collection in high-resolution respirometry. I think it is important for readers to understand that human samples are inherently variable, and this should be taken into account. I hope that this study precipitates others who use different types of biological samples to test the effect of potential residual rotenone effects.

Reviewer 2 ·

Basic reporting

The manuscript is clearly written and well structured. Raw data is appropriately shared.

The manuscript could benefit from more explicit descriptions of the various mitochondrial respiration states used in the SUIT protocols (i.e., CI leak, etc). While these states are familiar to those in the field, brief physiological definitions would be beneficial to those new to the field.

While the recommendation to exclude ROT in same-day experiments is well justified, the authors should elaborate on why it is often included in mitochondrial respiration studies and why its exclusion would not compromise evaluations of skeletal muscle mitochondrial respiration.

The authors should consider expanding their discussion to note that other commonly used SUIT inhibitors, such as oligomycin and antimycin A, both of which have potent effects on mitochondrial respiration, may also persist between runs if proper cleaning or overnight soaking is not performed. A brief comment is made about potential Antimycin A carryover in the discussion, but a broadened statement including all inhibitors and encouraging investigators to consider residual contamination of all inhibitors would be of benefit for the research community.

Figure 1 indicates that 17 participants were in Experiment four, while line 374 in the discussion states 18.

Experimental design

The study addresses an under-discussed and under-studied methodological issue regarding the use of inhibitors in high-resolution respirometry. The design logically builds across four experiments to assess this.

Methods are well-detailed and sufficient.

It would be helpful to explain how samples were assigned to chambers. Since contamination is chamber-specific, understanding whether assignment was randomized or counterbalanced would strengthen confidence in the experimental rigor.

Validity of the findings

The study presents well-analyzed data, with statistical methods clearly described and raw data and analysis code made publicly available.

The study clearly demonstrates that the recommended Oroboros wash protocol reduces residual ROT effects more effectively than simple rinsing. However, since measurable inhibition persists even after washing following standard Oroboros guidelines, it would be valuable to investigate whether further prolonging the wash steps beyond the current recommended standard could fully eliminate residual ROT contamination. Even exploratory data on this point would help guide best practices for future high-resolution respirometry experiments for studies where it is necessary to use Rotenone inhibition.

However, the inclusion of Experiment 4 requires some clarification. While the authors present Experiment 4 as reinforcing the conclusions of Experiment 3, the protocols differ not only in exercise design but also in the SUIT protocol, particularly the inclusion of octanoyl-carnitine to stimulate ETF-linked respiration. This change in substrate input alters the metabolic pathways being assessed, limiting the extent to which Experiment 4 can directly validate findings from Experiment 3 or prior literature reporting post-exercise declines in carbohydrate-linked respiration. Although the authors acknowledge these protocol differences, the claim that Experiment 4 offers “greater confidence” in the findings overstates its role in validating Experiment 3.

Reviewer 3 ·

Basic reporting

Taylor et al. address a highly relevant topic in their manuscript, reporting a potential misunderstanding in studies using high‐resolution respirometry during consecutive experiments with Oroboros equipment. Specifically, they assess this issue in oxygen‐consumption measurements in human muscle fibers, highlighting the difficulty of analyzing mitochondrial physiology in pre‐ and post‐exercise conditions. They demonstrate that the mitochondrial complex I inhibitor rotenone can contaminate the Oroboros chamber and affect the results of subsequent experiments. The authors provide examples of studies in which such misunderstandings may occur and discuss these in light of other mitochondrial parameters, such as citrate synthase activity. The importance of this work is clear: they show that the basic manufacturer's recommended cleaning procedures are insufficient to remove chamber contamination, and that residual inhibition can bias multiple experiments. However, I have a few questions and suggestions to help improve the quality of the paper.

Experimental design

In general, the paper is clear, well structured, and provides sufficient information about the procedures, including a helpful figure summarizing the experiments. The terminology is appropriate and easy to interpret. However, I have one concern regarding the experimental design. Although the authors present it in detail, my understanding is that the main issue stems from adding rotenone only to the MiR before introducing muscle fibers in most measurements, despite the washes. The sole exception was Experiment 3.

Validity of the findings

Considering this reviewer's criticism about the addition of rotenone to the medium before experiments without a biological sample, the effect of the inhibitor added exclusively to the medium may differ from that of the inhibitor added in the presence of biological samples. Because rotenone is hydrophobic, it can interact directly with tissues, and mitochondria‑rich samples may carry over residual inhibitor. Therefore, it is reasonable to expect that experiments following a pure rotenone addition will exhibit a more pronounced inhibitory effect than those conducted after a complete assay containing biological samples and all substrates. In addition, the Oroboros Wiki (https://wiki.oroboros.at/index.php/ISS_and_chamber_cleaning) recommends washing the chambers with cells, tissue homogenate, or isolated mitochondria immediately after experiments for approximately 30 min to promote inhibitor binding to mitochondria and subsequent removal of high‑affinity inhibitors. Thus, to this reviewer’s knowledge, experiments in which rotenone is added only to the medium prior to the experiment are insufficient to draw strong conclusions. It would also be important to include washes with mitochondria‑rich samples, in addition to ethanol rinses, before subsequent experiments, as already suggested by Oroboros. If inhibition persists, the impact of the present study would be even more significant.

Additional comments

I suggest a different title, since the authors basically assess rotenone inhibition.

Minor comments:

- Please, add references to the washing protocols of Oroboros
- Line 310: There is an error, please replace "permabilised" with "permeabilized".

---

## Round 0.2 · Minor Revisions

please address the remaining minor concerns of one reviewer.

**PeerJ Staff Note**: Please ensure that all review, editorial, and staff comments are addressed in a response letter and that any edits or clarifications mentioned in the letter are also inserted into the revised manuscript where appropriate.

**PeerJ Staff Note**: It is PeerJ policy that additional references suggested during the peer-review process should only be included if the authors agree that they are relevant and useful.

·

Basic reporting

The writing of the article is of a good standard.

Experimental design

Point have been clarified sufficiently now.

Validity of the findings

The manuscript methodology is transparent with data presented fully.

Reviewer 2 ·

Basic reporting

The authors have made the requested revisions. Specifically, they have provided more information on respiration states used in the SUIT protocol which will improve understanding with readers less familiar with the technique. Additional information on the usage of rotenone and broadening the information about potential contamination from other inhibitors has also been included. Altogether, these changes enhance the clarity of the manuscript.

Experimental design

The authors have provided a clear explanation of how chambers were assigned across conditions and I am satisfied with their response.

Validity of the findings

Concerns regarding the interpretation of Experiment 4 have been addressed and additional detail provided regarding it's relevance and inclusion in the manuscript.

Additional comments

The authors have addressed all major concerns raised in my initial review. I support acceptance of the manuscript in its current form.

Reviewer 3 ·

Basic reporting

I maintain my previous comments regarding the basic reporting. Taylor et al. address a relevant topic in their manuscript, highlighting a potential source of misunderstanding in studies using high-resolution respirometry with Oroboros equipment in consecutive experiments—specifically in the context of pre- and post-exercise measurements.
They have now implemented sentences which can clarify the reader about methods to clean Oroboros chamber.

Experimental design

I understand that the inclusion of Experiment 3 was intended 'to test the assertion that having mitochondrial-rich samples within the chamber would reduce the potential contamination of rotenone in subsequent experiments' (in the authors' own words). However, it is clear that the effect of rotenone, when added alone, was stronger—and even statistically significant in Figure 5D. I would expect a more thorough discussion, at the very least addressing this statement from the authors in response to my concerns: 'While Experiment 2 may overestimate the true effect (in terms of reduction in mitochondrial respiration values), it is unlikely that all available rotenone will be sequestered by tissue in the quantities commonly used (i.e., for skeletal muscle 0.5–3 mg), particularly when many studies use concentrations higher than we used within our study.

Validity of the findings

Further discussion and clarification of these points would be sufficient to adequately support and finalize the conclusions.

---

## Round 0.3 · accepted · Accept

Thanks for addressing all reviewers' comments!

Reviewer 3 ·

Basic reporting

All concerns previously raised have been satisfactorily addressed by the authors. Now I have no issues with this manuscript.

Experimental design

Methods are clear and well structured.

Validity of the findings

The findings are relevant and now well discussed.